# Ni-Al Bronze in Molten Carbonate Manufactured by LPBF: Effect of Porosity Design on Mechanical Properties and Oxidation

**DOI:** 10.3390/ma16103893

**Published:** 2023-05-22

**Authors:** Camila Arcos, Carolina Guerra, Jorge A. Ramos-Grez, Mamié Sancy

**Affiliations:** 1Departamento de Ingeniería Mecánica y Metalúrgica, Escuela de Ingeniería, Pontificia Universidad Católica de Chile, Santiago 7820436, Chile; cearcos@uc.cl (C.A.); caguerra2@uc.cl (C.G.); jramos@ing.puc.cl (J.A.R.-G.); 2MIGA Millennium Institute (ICN2021_023), Santiago 8320211, Chile; 3CIEN-UC, Pontificia Universidad Católica de Chile, Santiago 7820436, Chile; 4Escuela de Construcción Civil, Facultad de Ingeniería, Pontificia Universidad Católica de Chile, Santiago 7820436, Chile

**Keywords:** NAB alloy, fuel cell, porous anode, additive manufacturing, mechanical properties, corrosion

## Abstract

Fuel cell technology has developed due to diminishing dependence on fossil fuels and carbon footprint production. This work focuses on a nickel–aluminum bronze alloy as an anode produced by additive manufacturing as bulk and porous samples, studying the effect of designed porosity and thermal treatment on mechanical and chemical stability in molten carbonate (Li_2_CO_3_-K_2_CO_3_). Micrographs showed a typical morphology of the martensite phase for all samples in as-built conditions and a spheroid structure on the surface after the heat treatment, possibly revealing the formation of molten salt deposits and corrosion products. FE-SEM analysis of the bulk samples showed some pores with a diameter near 2–5 μm in the as-built condition, which varied between 100 and −1000 μm for the porous samples. After exposure, the cross-section images of porous samples revealed a film composed principally of Cu and Fe, Al, followed by a Ni-rich zone, whose thickness was approximately 1.5 µm, which depended on the porous design but was not influenced significantly by the heat treatment. Additionally, by incorporating porosity, the corrosion rate of NAB samples increased slightly.

## 1. Introduction

Global energy demand has increased significantly due to world population growth and the industrialization of developing economies [1]. Energy production has been based mainly on fossil-fuel energy [2], which has increased the global warming effect with the rise of greenhouse gases, such as carbon dioxide (${\mathrm{CO}}_{2}$), in the atmosphere [3].

Fuel cell technology has been explored as an excellent alternative for reducing the dependence on fossil fuels and carbon footprint production because the fuel cells use clean energy with a high conversion efficiency, also allowing ${\mathrm{CO}}_{2}$ capture [4,5]. Fuel cells are electrochemical devices that convert chemical energy to electrical energy [6]. Molten carbonate fuel cells (MCFCs) are one of the most rapidly developing high-temperature fuel cell technologies [4] that offer promising solutions to reduce the environmental impact of energy production by generating electricity without emitting harmful substances into the atmosphere. The main components are electrodes and an electrolyte membrane with high ionic conductivity [7]. The fuel frequently used is $H_{2}$, which is oxidized into the water at the anode, releasing electrons which are involved in the oxygen ($O_{2}$) reduction reaction at the cathodic side, forming carbonate ions (${\mathrm{CO}}_{3}^{2-}$) in MCFCs. The electrolyte separates the anode and the cathode, which corresponds to a porous ceramic matrix [3,4], such as a ceramic lithium aluminum oxide (${\mathrm{LiAlO}}_{2}$), embedding a mixture of the molten carbonate, as reported by Abdollahipour and Sayyaad [8]. The cathode and anode are porous materials, offering a sizeable geometric surface area or density of active sites for chemical reactions [6,9]. Although MCFC technology has existed for decades, some open challenges remain for a real breakthrough regarding durability, lifetime, and cost of core components [6,7].

Ni has been widely used in MCFCs for the past two decades due to its good electrochemical activity and low polarization losses. However, high temperatures produce a collapse of the anode due to the creep and sintering problems, decreasing the contact between the anode and electrolyte matrix. However, the high operating temperatures imply that using precious metals is unnecessary for anodes. Porous nickel–aluminum alloys (NiAl) are commonly used due to their low density, high melting point, excellent acid/alkali corrosion resistance, good oxidation resistance at elevated temperatures, and good charge transfer conductivity. The standard porosity of anodes varies between 50 and 65%, with a pore size of around 2–3 µm [6,9]. Anodes are affected by compressive and thermal stresses during the operation of MCFCs, which favor the creep deformation that decreases their porosity and electrochemical activity [10,11]. Even though this alloy has been shown to be ideal as an anode, some authors found that the loss of porosity during the MCFC’s operation influences the catalytic performance and efficiency of anodes [10,12,13]. Some alloying elements have been incorporated into NiAl anodes to improve their structural stability, such as titanium [14], chromium [15,16], and copper [17,18]. Studies have demonstrated that the anodes can be mechanically reinforced by adding a third alloying element to the NiAl or incorporating a hard metal oxide as a coating to block the dislocation movement and the sintering that can occur during the cell’s operation [14,19]. These ternary systems have interesting electrochemical properties, good ductility and stiffness, low creep resistance, and good corrosion resistance [14,20]. Nguyen et al. [15] incorporated chromium into the Ni-Al anode, finding the best creep strain with Ni-5 wt.% Al-10 wt.% Cr. However, the chromium addition did not improve the electrochemical performance. Kim et al. [21] analyzed the creep behavior of Ni-(4–7 wt.%) Ni_3_Al and Ni-5 wt.% Ni_3_Al-5 wt.% Cr anodes for MCFC, finding that the creep deformation decreased synergistically with the Ni_3_Al and Cr inclusion. Li et al. [17] investigated a Cu-Al anode as cast and porous materials, reporting a Cu-rich phase, a Ni-rich phase, and an intermetallic composed of both materials. In addition, the authors noted that the yield strength of porous alloys increased with decreased porosity and that the relationship between porosity and yield stress follows the Gibson–Ashby equation. Moreover, they determined that reducing the deformation temperature increased the yield strength for cast and porous alloys. Although Ni-Al anodes are widely used for MCFC applications, a loss of the anode’s porosity and corrosion in molten carbonate due to the high temperatures have been reported.

Adding copper (Cu) into Ni-Al alloy can reduce the cost of anode manufacturing, increasing thermal conductivity and mechanical resistance. Therefore, Cu-Ni-Al can be ideal for extreme conditions [17,22], such as an anode for MCFCs [17,19]. Moreover, Fe could be introduced to the alloy Cu-Ni-Al, manufacturing a commercial NAB (Cu-Al-Ni-Fe) alloy characterized by a good corrosion behavior in hostile environments, such as marine environments [23]. Indeed, some authors study the corrosion behavior of alloys such as 51Fe-24Cr-20Ni and 5Fe-23Cr-58Ni-8Mo in molten salts [24].

In this work, a new porous Cu-Al-Ni-Fe alloy was fabricated by laser powder bed fusion (LPBF) through selective laser melting (SLM), which allows for the fabrication of samples with different pore geometry at a laboratory scale [25]. This is unlike other techniques that use salts to generate arbitrary porosity, such as conventional powder metallurgy [26]. The samples were exposed to molten carbonate in an aerated medium to simulate the MCFC conditions to better understand the anode’s degradation during exposure, particularly its mechanical properties and corrosion resistance.

## 2. Materials and Methods

### 2.1. Sample Fabrication

Cu-11Al-5Ni-4Fe wt.% alloyed powders (<40 μm, CNPC powder group CO) were used to manufacture the anode samples through a selective laser melting (Concept Laser Mlab cusing 200R Ge machine, Boston, MA, USA) that was set to 30 μm and 80 μm for the layer thickness and hatch space. In contrast, the power and laser speed were kept at 180 W and 600 mm·s^−1^. Samples were fabricated with a cylindrical shape, constructed vertically over a copper platform with two lattice types with a length and diameter of 6 mm and 8 mm. After fabrication, samples were removed from the Cu platform using wire electrical discharge machining (W-EDM) at 100 V and 2.5 A. Table 1 shows the ID of samples and their characteristics.

Before exposure, the metal samples were polished by wet grinding with grit sandpaper from #400 to #4000 to reveal their microstructure and then polished with colloidal silica suspension. Some samples were thermally treated at 900 °C for 2 h and then cooled in a furnace to analyze the effect of annealing on the mechanical and microstructural behavior.

### 2.2. Morphological and Chemical Characterization

The apparent porosity was estimated using Archimedes’ method, based on the Standard Test ASTM C373-88 [27] and through image analysis using the open-source program ImageJ. The microstructure was revealed using 5 g of Fe_3_Cl, 10 mL of HCl, and 100 mL of distilled water for 10 s. Images were obtained using an optical microscope (OM), Olympus model GX41 (Olympus, Tokyo, Japan), and a field-emission scanning electron microscope (FE-SEM), QUANTA FEG 250 (FEI, Lausanne, Switzerland) [28]. The surface analysis was analyzed using X-ray Photoelectron Spectroscopy (XPS), employing a K-alpha photoelectron spectrometer (Thermo Scientific, Waltham, MA, USA). The alloy phases were characterized with X-ray diffraction (XRD) using Rigaku equipment (Rigaku, The Woodlands, TX, USA), MiniFlex 600, detector D/tex Ultra 2 High-Speed 1D, and equipped with a Cu Kα1 radiation source (λ = 1.54056 Å). The chemical composition depth profiles of the corrosion products were measured using glow-discharge optical emission spectroscopy (GD-OES, Spectruma GDA 750 HR, Spectruma, Hof, Germany).

### 2.3. Gravimetric Measurements

Anode samples were exposed to Li_2_CO_3_-K_2_CO_3_ molten salt at 550 °C for 21 days in aerated conditions, with eutectic composition (62:38 mol. %), as described by Cassir et al. [29], Ricca et al. [30], and Lair et al. [31]. Gravimetric measurements of samples were carried out in quadruplicates to ensure replicability. Every 7 days, samples were removed from the furnace and cleaned to eliminate the crystallized salt from the entire surface by using beakers filled with 25 mL of hot distilled water (~100 °C), which were placed in a sonicator bath (Elma D-78224 Singen/Htw, Singen, Germany) for 30 min for bulk samples and 1 h for porous samples. Subsequently, the samples were dried with hot air and weighed until they reached a constant value, as reported by the ASTM G1-03 [32]. The average mass (%) was calculated using Equation (1).
(1)$$\frac{m_{i}-m_{f}}{m_{i}}\times 100$$
where $m_{i}$ and $m_{f}$ are the initial and final sample masses at different exposure times.

The corrosion rate (CR) was estimated following the ASTM G1-03 [32] using the following equation:(2)$$\frac{\left(K\times W\right)}{\left(A\times T\times D\right)}$$
where *K* is a constant in the corrosion rate equation, *T* is the time of exposure in hours, *A* is the area in centimeters square (cm^2^), *W* is the mass loss in grams (g), and *D* is the density in grams per cubic centimeters (g·cm^−3^).

### 2.4. Mechanical Characterization

Before exposure, the microhardness of samples was evaluated using a micro-Vickers durometer in triplicate (Wilson^®^ VH1150 Macro Vickers Hardness Tester, Ontario, Canada) under 0.5 kgf of force, and the compression test was assessed following the ASTM E9 using an Instron 4200 machine with a speed test of 0.05 min^−1^.

## 3. Results and Discussion

### 3.1. Microstructural and Chemical Characterization

In this work, the apparent porosity and density of bulk (B) and porous samples (G) in as-built conditions were evaluated, determining a porosity near 3.5 ± 1.7% for the B sample, 81.4 ± 12% for the 1G sample, and 84.9 ± 11% for the 2G sample and a density close to 7.3 g·cm^−3^ for the B sample, 1.36 g·cm^−3^ for the 1G sample, and 1.15 g·cm^−3^ for the 2G sample.

Figure 1 shows the micrographs of the B and G samples in as-built conditions, revealing a martensite phase with an acicular shape, as shown in Figure 1a–c [33]. After the heat treatment, a spheroid structure was formed on the 1GHT and 2GHT samples, as described in annealing steels [34]. Therefore, it is possible to appreciate that after the heat treatment, the martensite disappears, and it is only possible to observe the grain form with a spheroid structure. Figure 1 reveals that the B, 1G, and 2G samples exhibit a microstructure composed of a darker matrix and precipitated phases in a lighter color [35,36], identifying the martensitic phase as β′, also known as “retained β” or as β′ phase [37]. The heat treatment influenced a microstructure change, as shown in Figure 1d,e. For example, the lighter zones were related to the α phase, which is a copper-rich solid solution [38], as reported by Tavares et al. [39], who stated that the heat treatment propitiated the β′ transformation to the equilibrium one [40], which can be attributed to the darker regions.

Figure 2 shows the FE-SEM images that reveal that the bulk sample (B) had a homogeneous surface with some pores whose diameter was near 2–5 μm (white arrow) formed possibly during fabrication, as shown in Figure 2a. In contrast, porous samples had a more heterogenous surface, with some unmelted spheric powders, sticks, and fissures, as seen in Figure 2b–e. After exposure to molten salt at 550 °C, all samples had a heterogeneous surface with inlays (lighter zones) and sharp or ridged geometric shapes (highlighted with black arrows), which could be molten salt deposits and corrosion products on the surface, as reported by Gupta and Mao [41]. Some authors have described the formation of different oxides on the metal surface when exposed to molten salts with Li content, such as LiAlO_2_ and LiFeO_2_, which can be formed due to the high reactivity of Al and Fe and reactive in oxidizing environments containing CO_3_ [24,42,43,44], which can diffuse to the surface [24,43,44]. It has also been reported that CuO, Cu_2_O, and NiO have low solubility in molten salt [42,43,45] and, thus, can be adhered to the metallic surface.

Figure 3 shows the EDS elemental mapping performed on the sample surface, revealing that the metal surface was composed mainly of Cu and Al for porous samples, 1G and 2G, before exposure. The Al content increased after heat treatment in the 1GHT and 2GHT samples, even more significantly than Cu, which could be related to the more negative reduction potential of alumina oxide in comparison to copper oxides, forming a thin alumina (Al_2_O_3_) film over the surface, as described previously by Hasegawa et al. [46]. The Cu content was more remarkable than the Fe, Al, and Ni content post-exposure.

In addition, FE-SEM analysis revealed local zones rich in Cu and Fe content and a more homogeneous distribution of Ni and Al. Tang et al. [44] studied the corrosion resistance of a Ni-10Cu-11Fe alloy as an anode exposed to molten carbonate in the presence of oxygen, determining that the formation of oxide was mainly composed of Fe and Cu. In addition, Spiegel et al. [47] reported the formation of magnetite (Fe_3_O_4_) and hematite (Fe_2_O_3_) on Fe-based alloys after exposure to Li_2_CO_3_-K_2_CO_3_ eutectic mixture, followed by the appearance of LiFeO_2_ or LiFe_5_O_8_, which are highly insoluble in the molten carbonate [48]. Audigié et al. [42] studied a nickel–aluminide coating exposed to a eutectic mixture of NaNO_3_-KNO_3_ and observed the presence of NiAl_2_O_4_ and Al_2_O_3_ oxides. De Miguel et al. [24] established the existence of aluminum-lithium oxide (LiAlO_2_) on the surface of an alloy due to the high reactivity of aluminum in molten salts in aerated environments, which improved the corrosion resistance of the material [42].

Figure 4 shows FE-SEM images and EDS analysis of a cross-section of bulk and porous samples after 21 days of exposure to the molten salt, revealing a homogenous distribution of the alloying elements on the alloy. For porous samples, a thin film was formed that was composed principally of Fe, then Al, followed by a Ni-rich zone, whose thickness was approximately 1.5 µm, which depended on the porous design. The heat treatment did not drastically influence the layer formed on these samples. However, a local zone in Cu content was determined [44,48,49]. It has been proposed that at the beginning of the exposure to the molten salt, Cu, Al, Fe, and Ni are oxidized, generating the oxide layer. Tang et al. [44] exposed a Ni10Cu11Fe alloy to Na_2_CO_3_-K_2_CO_3_ salts. The authors found an inner oxide layer composed mainly of NiFe_2_O_4_ that could be formed by the reaction between Fe_2_O_3_ and NiO, which is more stable than Fe_2_O_3_. In the NAB alloy, Fe is also the most reactive metal in the alloy [44] that can diffuse to the surface quickly. Therefore, an intermediate layer rich in Cu and Ni oxides (CuO, Cu_2_O, and NiO) can form. Goupil et al. [49] established that the Cu_65_Ni_20_Fe_15_ alloy used as an inert anode for aluminum electrolysis in oxygen presence can induce the formation of Fe_2_O_3_ precipitates and CuO, and then, Fe_2_O_4_ can be formed. Luo et al. [48] indicated that after a short exposure time, the corrosion products formed on the sample SS316L surfaces are mainly LiFeO_2_ with a small amount of Fe_3_O_4_, Fe_2_O_3_, and LiFe_5_O_8_ and an inner layer of NiO. Moreover, a thin alumina film (Al_2_O_3_) could form on the surface samples [46]. Therefore, all this oxide forms a layer that achieves stability over time.

Figure 5 shows a compositional depth profile obtained by GD-OES of the bulk sample before and after exposure, revealing the variation of each element (O, Al, Fe, Ni, and Cu) across the sample. Up to a depth of 1.5 µm, the oxygen had the highest mass concentration, which decreased drastically, which could be considered the oxide layer thickness. Cu was the second predominant element up to 1.5 µm, which increases until it reaches a stable value. Al was the third element, up to approximately 2.1 µm, reducing its content later. Fe, Al, and Ni have low quantities on the surface, which increased near 1.5 μm, demonstrating the transition from an oxide composed mainly of copper until it arrived at the base metal. According to the GD-OES, the oxide was formed primarily by Cu. For H samples, the inner layer was composed of other alloying elements. In particular, the B sample before exposure had a more significant amount of Fe on the surface, later Al and Ni. After exposure, Ni content increases in the inner layer.

Figure 6 shows the XRD patterns of bulk samples as-built conditions and after the heat treatment. The patterns show the presence of α-Cu (FCC), κ-phases (intermetallic), and the metastable β′-phase (martensite) in both conditions. The pattern of bulk samples in as-built conditions demonstrated thicker peaks and less smoothness than after heat treatment, which was attributed to internal defects due to rapid solidification [33]. After heat treatment, a higher amount of crystalline phases was determined, which can correspond to the equilibrium phases, such as the α-Cu that can reduce the residual stress and homogenizes the structure. Alkelae et al. [50] described the microstructure transition at different cooling rates for Cu-11Al-5Ni-4Fe samples (in wt.%), showing through a phase diagram that the β-phase (BCC) is stable at high temperatures (above 1050 °C). The cooling rate can favor the formation of intermetallic compounds, such as κ-phase, and solid solutions, such as α-phase (FCC). Therefore, the B sample could have good mechanical properties because the α-phase is a ductile and malleable phase that can reinforce the κ-phase. It is important to note that the martensite did not influence the XRD pattern after exposure because it is a metastable phase and decomposed in the stable phases at the test temperature.

### 3.2. Mechanical Properties

Figure 7 shows the microhardness of bulk and porous samples measured on the surface of the samples. The bulk sample exhibited a maximum value of 315.6 ± 15.8 HV, followed by the 2G sample with 304.1 ± 14.5 HV and the 1G sample with 279.7 ± 18.9 HV. After the heat treatment, the microhardness decreased by half, close to 164.2 ± 3.2 HV for the 2GHT sample and around 152.2 ± 10.9 HV for the 1GHT sample. According to Orzolek et al. [36], the microhardness decreases when the NAB alloy is heat-treated due to a microstructural transformation. For example, the martensitic microhardness is around 400 HV, while the α-phase value is near 320 HV. Lv et al. [51] determined that the microhardness in a NAB alloy was about 280 HV, which decreased to 245 HV after heat treatment at 675 °C for 2 h, which can be associated with a transformation of β′ phases into α. The B, 1G, and 2G samples had higher hardness, close to 300 HV, which was reduced to near 160 HV for the 1GHT and 2GHT samples after the heat treatment. After exposure, the micro-hardness was not registered in this work due to the corrosion products and deposits formed on the surface that were not soluble in hot water.

Figure 8 shows compression stress–strain curves of samples before and after exposure to the molten salt. The B sample exhibited the highest compressive strength and elongation due to its lower porosity [52]. The 1G porous sample presented a greater strength than the 2G sample, which has a similar porosity, around 84%, which in this case had the shape of a gyroid lattice [52] that defines its mechanical properties. It has been shown that outer shell wall inclusion in highly porous structures reinforces the alloy, improving the maximum compressive strength, and is the factor that most impacts the compressive mechanical properties [53].

Before exposure, porous samples were drastically influenced by the thermal treatment, revealing that annealing decreases the maximum strength and increases the elongation due to the residual stress release accumulation during the manufacturing process [40]. After 21 days of exposure to molten salt, the material suffered a drop in its compressive mechanical properties, especially for samples without heat treatment, reaching a maximum strength similar to that of the heat-treated samples. The latter can be attributed to the prolonged exposure of samples to 550 °C, which modified their internal microstructure, fostering their similarities. Therefore, the maximum strength diminished in all samples, but the elongation increased, which can be related to grains coarsening [54] during the exposure, since the temperature provided enough energy to increase the grain size [40], according to the phase diagram [50]. Additionally, the post-processing of the samples is unnecessary if the alloy is exposed to high temperatures, and thus, it can be used in its as-built condition.

Wee et al. [54] reported the creep behavior for porous anodes in fuel, varying the load between 0.1 and 0.7 MPa and with different temperatures. The Ni-Cr and Ni-Al anodes revealed an adequate creep rate between 5 and 10% creep strain at 100 h. The bulk NAB alloy achieved a 0.01% creep rate after 1000 h using stress of 3.1 MPa at 550 °C [55], which is promising for its use as an alloy in MCFCs. Table 2 summarizes the effect of the heat treatment on the maximum strength and elongation of the NAB samples, as bulk and gyroid, extracted from the compressive strain curves. The porous samples decreased the maximum of stress in comparison to that of the bulk samples, which was more drastic after exposure. However, the elongation was not significantly influenced by the porosity and exposure.

### 3.3. Gravimetric Measurements

Figure 9 shows the mass variation (ΔW) and corrosion rate (CR) of NAB samples as functions of the exposure time in molten carbonate at 550 °C in aerated conditions. The B and 2G samples lose mass for a longer exposure time, and the mass quantity loss also increases, revealing a directly exponential relationship. Otherwise, the masses of the 1G, 1GHT, and 2GHT porous samples increased over time. It is important to note that 48 h could be established as a critical time, since for all samples, the corrosion rate becomes a constant value. The molten salt deposits and corrosion products formed on the surface could cause the mass gain of 1G, 1GHT, and 2GHT samples. The corrosion products can be formed by the interaction of the molten carbonate with the alloy, such as LiFeO_2_, or by the presence of oxygen, forming Al_2_O_3_, which has been demonstrated to be adherent to the metallic surface.

Audigié et al. [42] studied the corrosive behavior of aluminide and nickel–aluminide coatings in molten salts, suggesting that the aluminide coatings had a higher weight gain due to their protective effect. Therefore, the heat-treated samples (1GHT and 2GHT) had previously developed an alumina layer before exposure. Furthermore, De Miguel et al. [24] studied the corrosion behavior of the bulk alloy 51Fe-24Cr-20Ni exposed to molten carbonate at 700 °C. The authors proposed that the estimated mass loss was attributed to the lower protective properties of the oxide layer, which can also be associated with the solubility of the oxides in molten carbonates, as observed in the B and 1GHT samples, which lose mass through the exposure. Gomez-Vidal et al. [56] estimated the corrosion/protection rate according to the weight variation in alloys exposed to Na_2_CO_3_–K_2_CO_3_ at 750 °C, determining that it was 1080 ± 40 µm·y^−1^ for In800H (30 wt.% of Ni) and 4640 ± 40 µm·y^−1^ for SS321 (9 wt.% of Ni). The authors [57] indicated that the corrosion rate was reduced, while the amount of Ni increased in the alloy. In this work, the corrosion rates were 0.1973 mm·y^−1^ or 197.3 µm·y^−1^ for the B sample and 0.3742 mm·y^−1^ or 374.2 µm·y^−1^ for the 2G sample, which are lower than those obtained in In800H and SS321.

On the other hand, the mass gain or the positive weight is attributed to the corrosion products attached to the surface as CuO, Al_2_O_3_, and Fe_2_O_3_. According to Audigié et al. [42], Al_2_O_3_ has better adhesion to the surface. In addition, the pore geometry of the 1G samples could have made it more difficult for corrosion products to detach from the sample.

Figure 10 shows a physical model of the effect of the heat treatment on the corrosion product’s evolution over the surface. Before exposure, copper oxide formed clusters on all sample surfaces. In addition, the alumina oxide was generated as a cluster on the sample surface without heat treatment but as an outer layer on the sample surface with heat treatment. After exposure, the heat treatment also determined the corrosion products, forming an iron oxide caused possibly by the molten salt and metal reaction, under copper oxide, competing with aluminum oxide. The EDS analysis suggests the diffusion of iron to the surface after 21 days of exposure at high temperatures and aerated conditions, as seen in Figure 3 and Figure 4, which simulates a real medium for an anode of exposure to this kind of anode. In addition, a greater amount of copper oxide was formed in heat-treated samples.

Figure 11 shows the XPS spectrum surveys of Al, Fe, and Cu, analyzed on surface samples before and after exposure to molten salt, with and without heat treatment. As mentioned above, the heat-treated samples formed an oxide composed of Al and Fe over the surface, which was attributed to the high reactivity of Al and Fe with the O produced in the open-to-atmosphere furnace during the heat treatment. Additionally, the heat-treated samples showed a peak of around 74 eV, which could be related to the intermetallic compounds of AlNi, which is a κ_III_ stable phase [56]. It is explained that the annealing provokes the martensite decomposition into the equilibrium phases. The Cu XPS survey is consistent with and without heat treatment before and after exposure because Cu is naturally formed on the surface in the most significant proportion. It should be noted that lithium compounds were searched for in the XPS spectrum but were not found. Karfidov et al. [58] found similar results for the corrosion resistance of Monel 404 alloys (Cu-50Ni) and Hastelloy exposed to LiF–NaF–KF at 550 °C after 100 h and determined an outstanding corrosion resistance of Cu-Ni alloy under the studied conditions. Table 3 summarizes the changes in quantity by element produced before and after exposure. As shown, all elements decreased in the count after exposure due to the increase in O content, which was related to the oxide formed on the metal surface. The more considerable differences are in the aluminum survey, which was possibly present in the inner oxide layer, but after exposure, it is covered by iron oxide, as observed using EDS analysis. The XPS confirms an increase in the oxide amount after exposure, mainly composed of Cu, with a low amount of the other alloying elements. It has been proposed that Cu_2_O is a kind of copper oxide that protects against corrosion in aqueous media [59], which is present in the alloy before and after exposure but in a lower amount after exposure.

## 4. Conclusions

Bulk and porous samples were manufactured using LPBF as an anode for molten carbonate fuel cell application. The effect of annealing heat treatment and porosity on mechanical and corrosion performances was investigated. The following conclusions can be drawn from the research.

The porosity significantly influenced the mechanical response, decreasing the maximum strength by 27% when the porosity was more than 80%. The differences fall if the porous samples present an external wall, which contributes to the strengthening of the material. Therefore, a control in density gradient could increase the maximum strength of porous samples.Microstructurally, the bulk sample in its AB condition shows a great amount of martensite phase due to the quick solidification, which decreases in quantity after annealing, demonstrating a more significant amount of α-phase. During exposure, the phase composition had no effect on mechanical properties since the immersion temperatures contribute to phase homogenization, and these exhibited similar mechanical properties after exposure. From a corrosion point of view, was no evidence of preferential phase corrosion.The heat treatment (annealing) performed before exposure produced a thin oxide layer composed principally of Al, which protected the alloy from further corrosion in contact with molten salt. However, it is unknown if the layer can influence its anode functions negatively due to the isolation that it can produce, for which a catalytic study is recommended.The molten salt exposure influenced the formation of the corrosion products, composed principally of Cu, Al, and Fe, whose thickness, determined using GD-OES analysis, was approximately 1.5 µm. The 2G sample had the worst performance since it lost mass and had a corrosion rate of 0.37 mm·y^−1^, followed by the bulk sample that had a corrosion rate of 0.19 mm·y^−1^. Therefore, it is possible to conclude that the pores and their geometry affected the corrosion, since the oxides formed on the surface of these samples were not sufficiently protective or adherent.Regarding the suitability of Ni-Al bronze alloy fabricated through additive manufacturing as an anode for molten salt carbonate, it could be a great candidate due to its low corrosion rate compared to its counterpart and its high strength under compressive loads.

## Figures and Tables

**Figure 1 materials-16-03893-f001:**
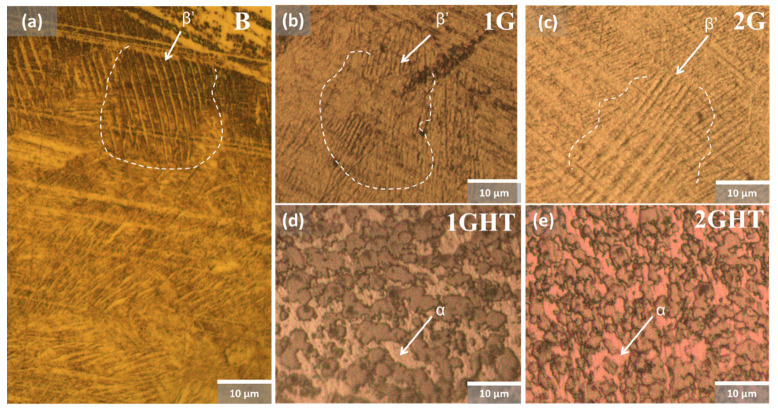
Micrographs of (**a**) bulk, (**b**,**d**) gyroid with wall structure, and (**c**,**e**) gyroid structure. Samples (**b**,**c**) without and (**d**,**e**) with thermal treatment.

**Figure 2 materials-16-03893-f002:**
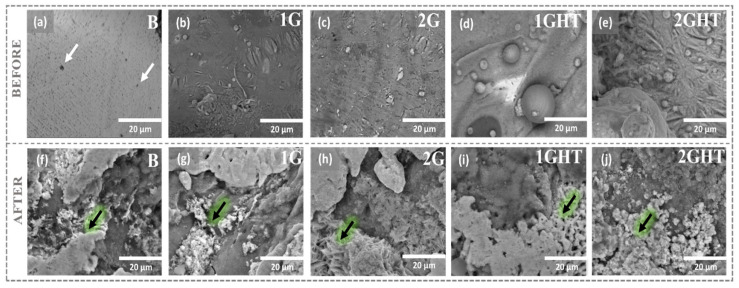
FE-SEM images of (**a**,**f**) bulk, (**b**,**d**,**g**,**i**) gyroid with wall structure, and (**c**,**e**,**h**,**j**) gyroid structure (**a**–**e**) before and (**f**–**j**) after exposure to molten salt. Samples (**a**–**c**,**f**–**h**) without and (**d**,**e**,**i**,**j**) with thermal treatment.

**Figure 3 materials-16-03893-f003:**
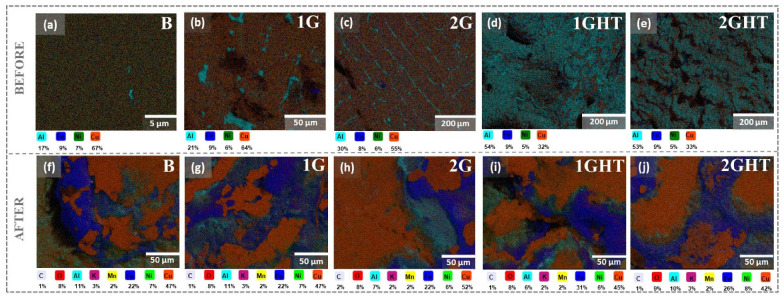
EDS surface mapping of (**a**,**f**) bulk, (**b**,**d**,**g**,**i**) gyroid with wall structure, and (**c**,**e**,**h**,**j**) gyroid structure, (**a**–**e**) before and (**f**–**j**) after exposure to molten salt. Samples (**a**–**c**,**f**–**h**) without and (**d**–**e**,**i**–**j**) with thermal treatment.

**Figure 4 materials-16-03893-f004:**
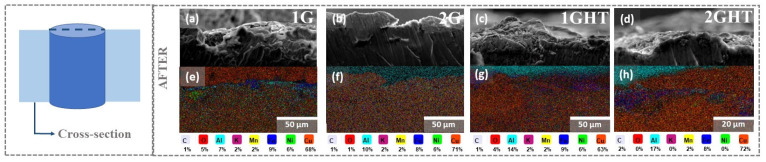
(**a**–**d**) FE-SEM and (**e**–**h**) EDS analysis of the cross-section of (**a**,**b**) gyroid with wall structure and (**c**,**d**) gyroid structure after 21 days of exposure to molten salt. Samples (**a**,**b**) without and (**c**,**d**) with heat treatment.

**Figure 5 materials-16-03893-f005:**
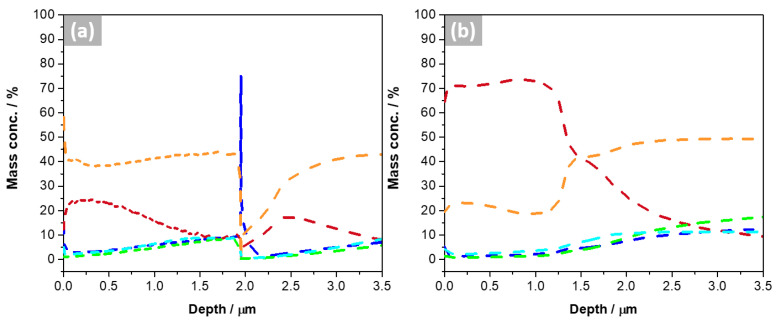
GD-OES of bulk samples (**a**) before and (**b**) after 21 days of exposure to molten salt. (--) O, (--) Al, (--) Fe, (--) Ni, and (--) Cu.

**Figure 6 materials-16-03893-f006:**
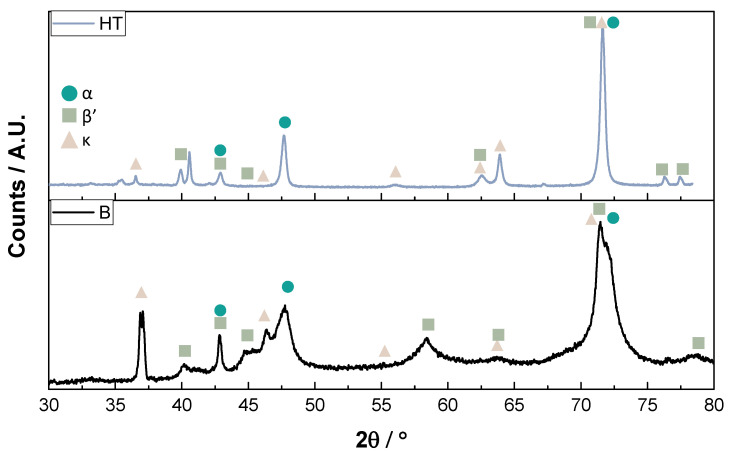
XRD patterns of the bulk sample in as-built condition and after heat treatment (HT).

**Figure 7 materials-16-03893-f007:**
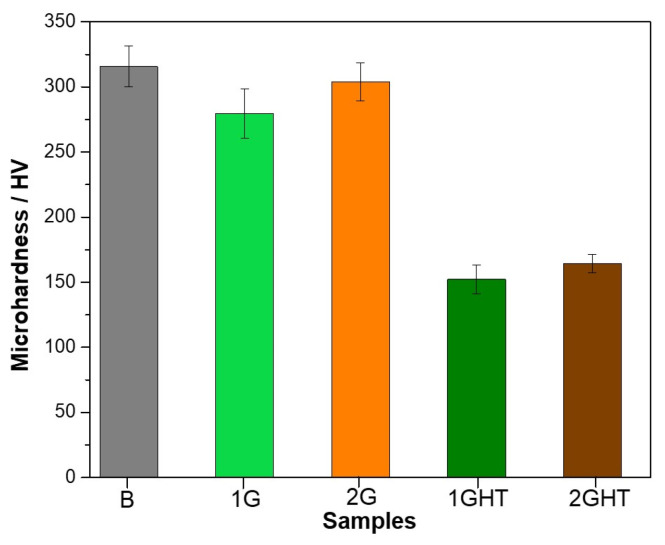
Microhardness (HV) of the samples before exposure.

**Figure 8 materials-16-03893-f008:**
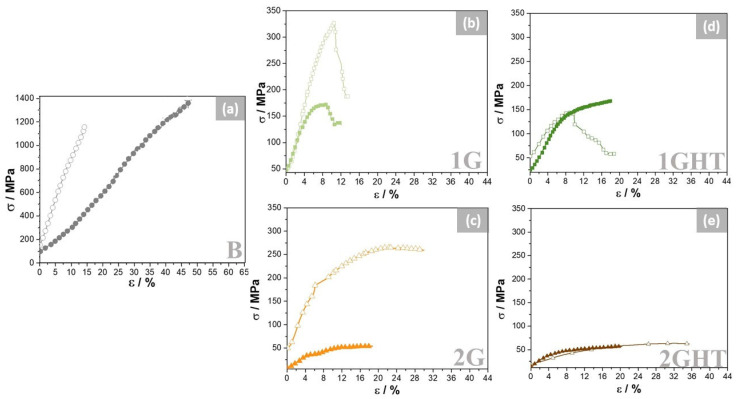
Compression stress–strain curves of (**a**) bulk, (**b**,**d**) gyroid with wall structure, and (**c**,**e**) gyroid structure. Samples with “empty symbols” before and with “filled symbols” after 21 days of exposure to molten salt.

**Figure 9 materials-16-03893-f009:**
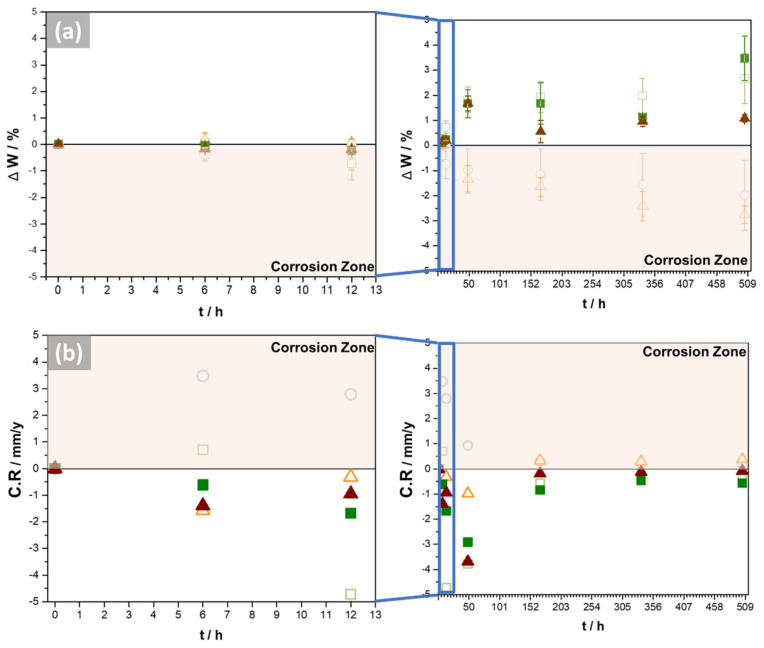
The variation of weight (**a**) and corrosion rate of samples (**b**) after 21 days (504 h) of exposure to molten salt. The samples exposed are 
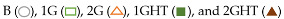
.

**Figure 10 materials-16-03893-f010:**
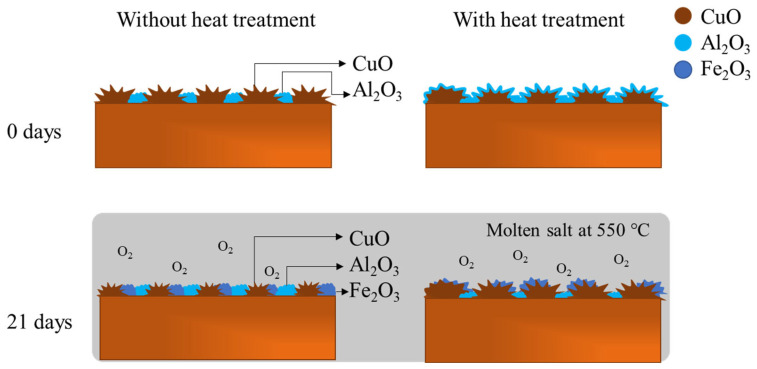
Physical model of the corrosion product evolution as a function of exposure time in molten salt at 550 °C in aerated conditions.

**Figure 11 materials-16-03893-f011:**
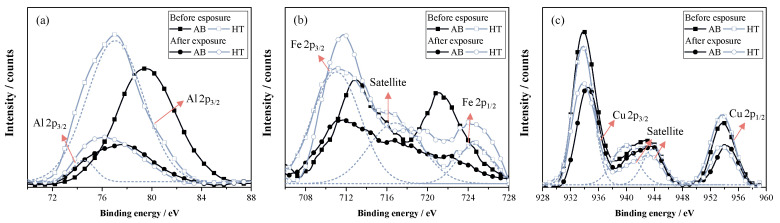
XPS survey of sample in as-built condition and heat treated before and after exposure. (**a**) Al, (**b**) Fe, and (**c**) Cu.

**Table 1 materials-16-03893-t001:** ID for NAB samples.

ID	Lattice Type	Heat Treated	Image
B	Bulk	No	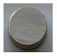
1G	Gyroid + wall	No	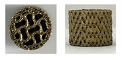
1GHT	Yes
2G	Gyroid	No	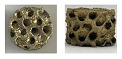
2GHT	Yes

**Table 2 materials-16-03893-t002:** Summary of compressive strain at room temperature tested before and after 21 days of exposure.

Sample ID	Before Exposure			After Exposure		
	σ_UTS_ (MPa)	Max. ε (%)	% Var. *	σ_UTS_ (MPa)	Max. ε (%)	% Var. *
B	1140.9 ± 27.1	20.8 ± 0.5	100	1415.5	59.9	100
1G	317.7 ± 10.0	19.0 ± 0.5	27.8	177.4 ± 6.5	19.1	12.5
2G	138.1 ± 6.0	19.5 ± 5.9	12.1	50.6 ± 5.7	24.6 ± 4.9	3.6
1GHT	237.7 ± 5.6	37.3 ± 0.2	20.8	168.7	29.2 ± 5.3	11.9
2GHT	96.8 ± 33.0	51.8 ± 10.4	8.5	64.9 ± 10.8	26.9 ± 1.5	4.6

* Percentage or proportion of the maximum stress achieved of porous samples concerning bulk.

**Table 3 materials-16-03893-t003:** Summary of chemical relative quantity of surface alloy elements obtained using XPS.

Elem.	Al				Fe				Cu			
Peak	74 eV		76 eV		712 eV		724 eV		933 eV		954 eV	
Exp.	AB	HT	AB	HT	AB	HT	AB	HT	AB	HT	AB	HT
Before	0	48.9	684.1	789.5	232.8	339.7	213.6	85.6	1145.3	1291.6	680.2	734.4
After	0	142.6	279.6	393.3	131.1	335.0	27.0	116.8	954.3	1067.6	409.5	430.0

## Data Availability

The data is unavailable due to privacy or ethical restrictions.

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
