# Peer review of "Ni-Al Bronze in Molten Carbonate Manufactured by LPBF: Effect of Porosity Design on Mechanical Properties and Oxidation"

_materials, 2023, doi:10.3390/ma16103893_

Round 1

Reviewer 1 Report

In this paper, the authors studies effect of porosity design on mechanical properties and oxidation of a Ni-Al bronze. 

Authors should be careful with subscripts throughout the text.

Parts of the template appear incorporated into the manuscript. Like for example the first line of the introduction.

Based on the introduction and the objective of the article, applications of these new materials should appear in order to meet the true objectives. If the manuscript is based on the synthesis and characterization process, the introduction should be rewritten.

The materials and methods section it is not clear, how you prepared the samples, what reagents did you use?

Figure 4 is really difficult to interpret, you should perform the analysis closer and with better resolution.

The manuscript has a lot of bibliographic information, looking more like a review than a research article. In turn, in terms of research, it seems more like a work of determination than a research process.

Finally, the conclusions of the article are poor. It must show more detailed information about the work, It simply justifies the results and does not comment on what innovation it brings.

Therefore, I think that this paper needs many changes for its publication, so I do not recommend accepting it.

Author Response

Dear reviewer, please see the attachment.

Kind regards, 

Reviewer 2 Report

The authors presented quite interesting data on the study of the mechanical and corrosion properties of Ni-Al bronze. After reading the text of the manuscript, the reviewer had some questions.

1. In the text of the entire manuscript, the reviewer did not find an explanation of what the LPBF method is.

2. You should delete the first sentence from the Introduction.

3. What is the reason for the choice of the quadruple system? How do the authors explain the choice of Fe as one of the additives?

4. Check the correct notation in formula (2).

5. In Fig. 7: Specify the x-axis or present the data as a histogram.

Answers to these comments will help to better understand the idea of the authors.

Author Response

(The authors gave the same response as above.)

Reviewer 3 Report

The work focuses on research and development of fuel cell technology, i.e. on one of the areas aimed at reducing the wear of collapsible cells, which significantly affect the central pollution. The research focuses on the Cu-11Al-5Ni-4Fe alloy processed into anodes with the use of additive method. The article has a thematic scope, research results and conclusions appropriate as for scientific articles.

 Minor editing fixes:

 - in Figs. 3 and 4, there is a barely visible legend

 - Fig. 8 hardly visible scales on the axes.

 In their current form, they are not visible on a paper printout.

As for the hardness measurements methodology, the value of the force used during the tests and the related microhardness scale were not indicated.

Author Response

(The authors gave the same response as above.)

Round 2

Reviewer 1 Report

In this paper, the authors studies effect of porosity design on mechanical properties and oxidation of a Ni-Al bronze. The authors use different techniques for its determination. Also, the authors have responsed to my questions and they have improved the paper. Although, certain parts of the paper should be better reviewed:

There are still typographical errors in the manuscript, mainly in the subscripts. Check the bibliography section and the abstract.

I still thinks that too many references are used in the manuscript, making it mistaken for a bibliographical review.

Even so, the article is well ordered and after looking at the paper carefully, I consider that this paper need a minor revision for publication.

Author Response

Thank you for your comment. As suggested, we have checked and corrected the mistakes in the subscript and reduced the number of references to improve the revised manuscript's quality.

Reviewer 2 Report

Thanks to the authors for the clarification. The text of the manuscript has become more readable.

Author Response

We appreciate your opinion and careful manuscript revision, which allowed us to improve the revised manuscript.